# Robotic Weld Image Enhancement Based on Improved Bilateral Filtering and CLAHE Algorithm

Peng Lu [1] and Qingjiu Huang [1,2,*]

1 School of Information and Electronic Engineering, Zhejiang Gongshang University, Hangzhou 310018, China
2 Control System Laboratory, Graduate School of Engineering, Kogakuin University, Tokyo 163-8677, Japan
* Correspondence: huang@cc.kogakuin.ac.jp

**Abstract:** Robotic welding requires a higher weld image resolution for easy weld identification; however, the higher the resolution, the higher the cost. Therefore, this paper proposes an improved CLAHE algorithm, which can not only effectively denoise and retain edge information but also improve the contrast of images. First, an improved bilateral filtering algorithm is used to process high-resolution images to remove noise while preserving edge details. Then, the CLAHE (Contrast Limited Adaptive Histogram Equalization) algorithm and Gaussian masking algorithm are used to enhance the enhanced image, and then differential processing is used to reduce the noise in the two images, while preserving the details of the image, enhancing the image contrast, and obtaining the final enhanced image. Finally, the effectiveness of the algorithm is verified by comparing the peak signal-to-noise ratio and structural similarity with other algorithms.

**Keywords:** robot welding; improved bilateral filtering; improved CLAHE algorithm

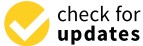



## 1. Introduction

Due to the rapid development of modern industry, the use of robots for welding has become an indispensable transformation for processing enterprises in the field of industrial manufacturing. Current a commonly used robot in factory production is the teaching robot. The principle is that we first provide a fixed trajectory and then let the robot accurately implement the trajectory [1,2]. Many small and medium-sized machining enterprises themselves have no requirements for the accuracy and surface-roughness of the processed products, but the weld position often changes. Therefore, a vision sensor on the robot arm is usually installed to facilitate and accurately find the welding position, so as to achieve the purpose of automatic welding [3], that is, automatic welding based on machine vision [4].

Machine vision has a wide range of applications in the field of robotic automatic welding [5]. However, for most small and medium-sized enterprises, high-resolution industrial cameras are required to achieve welding accuracy. Due to the high cost and difficulty of purchasing such cameras, the demand for weld image enhancement technology is increasing. For example, in the marine anchor chain manufacturing industry, there is no requirement for the machining accuracy of the anchor chain, nor does it need to be polished. When the robot welds the anchor chain rung, the image captured by the camera will have noise and fine lines, yielding no obvious value for characterizing the weld. Therefore, it is necessary to enhance the weld image before looking at the weld, and use the low pixel camera to obtain the same weld recognition accuracy as a high pixel camera.

The CLAHE (Contrast Limited Adaptive Histogram Equalization) algorithm is an adaptive histogram equalization algorithm used until now to limit the contrast [6,7], which can limit the excessive improvement of image contrast. Compared with the ordinary AHE (Adaptive Histogram Equalization) algorithm, this algorithm uses the contrast restriction while performing histogram equalization for each sub block, which leads to the considerable complexity of the algorithm [8,9]. For the improved CLAHE algorithm, fusion

homomorphic filtering and CLAHE algorithms at home and abroad have been proposed for image enhancement of cloth defect points [10,11], but the detection effect at the edge of the image is not good. A bilinear interpolation CLAHE algorithm was proposed to improve the speed of the algorithm [12,13]; however, among other things, image quality was lost. An infrared image contrast enhancement algorithm with bilateral filter transformation combined with restricted histogram equalization algorithm was proposed to extend the dynamic range and enhance the detail of infrared images [14]. A CLAHE image enhancement algorithm based on adaptive brightness adjustment has been proposed. The algorithm first converts the RGB space of the image into an HSV space, extracts the brightness component of the image, and then adaptively adjusts the overall brightness of the RGB channel image according to the brightness value of the image, and finally applies the CLAHE algorithm to achieve image enhancement [15]. An improved CLAHE algorithm has been proposed to transform the image from RGB space to HSV space, limiting only the luminance component to contrast adaptive histogram equalization enhancement, then grayscale transformation of the original image, then histogram equalization of the transformed image, then normalization of the image, and finally RGB output [16]. An improved Retinex algorithm was proposed to improve the quality of low-illumination images, improve contrast, and reduce noise; the algorithm uses a guided filter for multi-scale Retinex algorithm in the brightness component, and performs adaptive saturation stretching in the saturation component. Then, the images are processed by histogram equalization, and the two algorithms are fused to improve the accuracy [17].

Based on the above background, this paper first proposes an improved bilateral filtering algorithm that not only eliminates fine lines and noise in the weld image, but also further improves the edge characteristics of the weld image. On this basis, the resulting image is enhanced by Gaussian mask processing and the CLAHE algorithm, and then differential processing is carried out to reduce the secondary noise in the image and retain the detail information of the image, thereby enhancing the contrast of the image and obtaining the final enhanced image. In addition, by comparing the peak signal-to-noise ratio and structural similarity of each algorithm with the weld image, this paper demonstrates the effectiveness of the enhancement algorithm proposed in this paper.

The structure of this paper is as follows. The second part is an introduction to the algorithm flow in this paper. The third section focuses on an improved CLAHE algorithm to improve the quality of weld images. The fourth section is the experimental results, which mainly analyze the feasibility of the experiment in this paper through subjective evaluation and objective evaluation. The fifth section is a conclusion.

## 2. Algorithmic Flow

The research in this paper is based on horizontal gear robot welding anchor chains, as shown in Figure 1. Figures 2 and 3 are 1.25 megapixel and 200,000 pixel raw images, respectively. This paper first performs an improved bilateral filtering of the original image to obtain a denoised image that preserves edge information. Then, the noise reduction image is enhanced by Gaussian mask processing and the CLAHE algorithm respectively, and the enhanced images D1 and D2 are obtained; the grayscale of D1 and D2 is subtracted to the high-frequency information image D3, and then the gray value of the high-frequency image and the CLAHE algorithm image is added, and the final detailed information is a clearer high-contrast HR image. The algorithm flow is shown in Figure 4.

Anchor chain links    welding seam    welding torch    camera    Robot body

**Figure 1.** Anchor chain horizontal gear robot welding device.

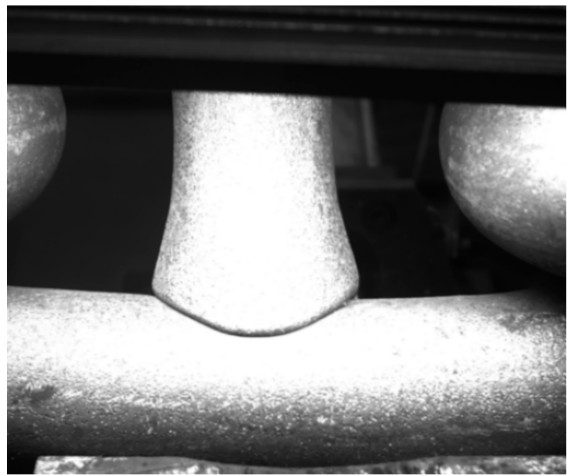

**Figure 2.** Original image 1 of a 1.25 megapixel anchor chain profile.

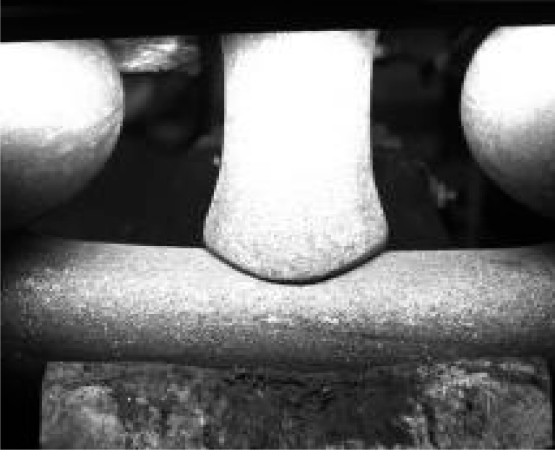

**Figure 3.** Original image 2 of a 200,000 pixels anchor chain profile.

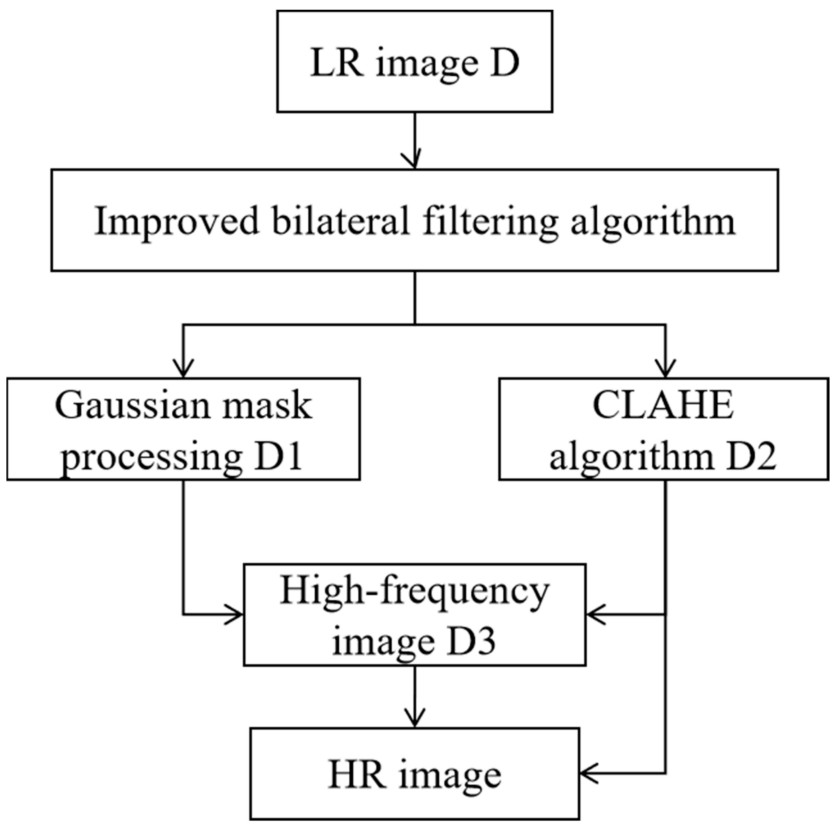

**Figure 4.** Algorithm flowchart.

### 3. Improved CLAHE Algorithm

*3.1. Improved Bilateral Filtering*

When filtering an image with bilateral filtering—not only the proximity of space, but also the similarity of grayscale—a smooth image is obtained by a nonlinear combination of the two [18–23]. The expression is shown in (1):

$$\hat{I}(x,y) = \frac{\sum_{(i,j)\in M_{x,y}} \omega_s(i,j)\omega_r(i,j)I(i,j)}{\sum_{(i,j)\in M_{x,y}} \omega_s(i,j)\omega_r(i,j)} \tag{1}$$

$$\omega_s(i,j) = exp(-\frac{|i-x|_2 + |i-y|^2}{2\delta_s^2}) \tag{2}$$

$$\omega_r(i,j) = exp(-\frac{\left|I(i-x)^2 - I(i-y)\right|^2}{2\delta_r^2}) \tag{3}$$

where $\hat{I}(x,y)$ is the filtered image; $M_{x,y}$ represents a set of $(2N+1) \times (2N+1)$ spatial neighborhood pixels centered on *(x,y)*; $I(x,y)$ represents the center point pixel value of $M_{x,y}$; $I(i,j)$ represents the pixel value at *(i,j)* in $M_{x,y}$; $\omega_s(i,j)$ is the spatial proximity factor; $\omega_r(i,j)$ is the grayness similarity factor; and $\delta_s$ and $\delta_r$ are filter parameters. Supposing $\omega(i,j)$ is the weight factor, then:

$$\omega(i,j) = \frac{\omega_s(i,j)\omega_r(i,j)}{c} \tag{4}$$

where *c* is constant. As can be seen from Equation (4), the weight factor is controlled by both the spatial proximity factor and the grayscale similarity factor, the spatial proximity factor decreases with the increase in the spatial position of the pixel and the center pixel, and the grayness similarity factor increases with the decrease of the gray value difference

of the two pixels. At the same time, when $\delta_s$ becomes larger, the number of pixels involved in weighting increases, and the image becomes blurry, but due to the limitation of $\delta_r$, the edge features are maintained. Therefore, bilateral filtering protects the edges of the image while denoising.

This article re-improves the compensation function based on the similarity of pixel gray levels within the window, as shown in the following equation:

$$\omega_r(i,j) = exp(-\frac{\left| I(i-x)^2 - I(i-y) - \tau(x,y) \right|^2}{2\delta_r^2})$$ (5)

wherein the compensation function $\tau(x,y)$ is set as follows:

(1) Judge the similarity of the gray value of the pixel and the center point in the filter window based on the similarity. If the absolute value of the difference between the pixel and the center point pixel is less than $\delta_r/3$, it is judged that $I(i,j)$ is similar to $I(x,y)$, and the original value of $I(i,j)$ is retained; otherwise, $I(i,j)$ is 0;

(2) Set the compensation function according to the number of similar points in the window. If the number of window pixels placed at 0 is less than 1/3 of the number of window pixels, set $\tau(x,y) = 0$; otherwise, follow step (3) to set it up;

(3) Introduce variables *Min*, *Max*, and *Mean*, which represent the minimum, maximum, and mean values of pixels in the filter window, respectively. Order $\varphi = I(x,y) - Mean$. If $\varphi > 0$, $\tau(x,y) = Max - I(x,y)$; if $\varphi < 0$, $\tau(x,y) = Min - I(x,y)$; if $\varphi = 0$, $\tau(x,y) = 0$.

### 3.2. Improved CLAHE Algorithm

After obtaining a high-resolution image with 3.1 denoising to preserve edge details, we combine the two techniques of adaptive histogram equalization and contrast limiting, namely the limit contrast adaptive histogram equalization (CLAHE) algorithm [24]. However, the algorithm only improves image contrast and has no noticeable effect in terms of image detail. The improved CLAHE algorithm is proposed below, which performs secondary noise reduction on the enhanced image, and highlights the details of the image while reducing noise to obtain the final enhanced image.

The CLAHE algorithm uses the histogram of each subblock to limit the amplitude of each subblock on the basis of the HE algorithm [25–27], thereby suppressing noise amplification and excessive local contrast enhancement. First of all, the image is divided into several subblocks, and then the histogram is cropped for each subblock; then, the histogram equalization adjustment is performed on each subblock, and finally according to the different positions of the subblocks, different interpolation operations are taken to obtain the transformed gray value, so as to achieve the limit contrast adaptive histogram equalization, the specific steps are as follows:

(1) The image is divided into continuous, non-overlapping subblocks of $m \times n$, the values of m and n can be 4, 6, 8, 16, etc., and each subblock contains the number of pixels N.

(2) The segmented subblock is processed with a Gaussian mask to obtain an image $H_1(x)$ after secondary noise reduction.

(3) Histogram equilibrium is performed on all subblocks obtained after splitting in step (1) to obtain its grayscale histogram, represented by $h(x)$.

(4) Calculate its clipping amplitude $T$:

$$T = C * \frac{N_x * N_y}{M}$$ (6)

where $C$ is the clipping coefficient, $N_x$ and $N_y$ are the number of pixels in the lower x and $y$ directions for each subblock, respectively, and $M$ is the gray level of the subblock.

(5) Crop the grayscale histogram and redistribute the image pixels. Each subblock histogram $h(x)$ is cropped according to the amplitude $T$ of the crop, and the pixels of

the cropped part are reassigned to each gray level with the number of gray levels *M*. We set the total number of pixels beyond the crop amplitude *T* to *S*, and the pixels reassigned at each gray level to *K*, to obtain Equations (7) and (8).

$$S = \sum_{x=0}^{M-1} \{max[h(x) - T]\} \tag{7}$$

$$K = \frac{S}{M} \tag{8}$$

The formula for the histogram after the reassignment of pixels is represented by *h(x)*, as shown below.

$$h_2(x) = \begin{cases} T + K, h(x) \geq K; \\ h(x) + K, h(x) < K; \end{cases} \tag{9}$$

(6) As shown in Figure 5, the tile area of the original image is uniformized and adjusted, and the mapping relationship between the image pixels and the grayscale conversion function of the tile area is used to perform interpolation operations to solve the gray value of the corresponding pixels at the edge points of the tile area, and the calculation efficiency can be improved. Depending on the number of neighbors, bilinear interpolation is used when the change function is four reference points. When the change function is two points, single-linear interpolation is used. When the change function is a reference point, the gray value of the block is used. The calculation process is as follows, and the enhanced image $H_2(x)$ is obtained.

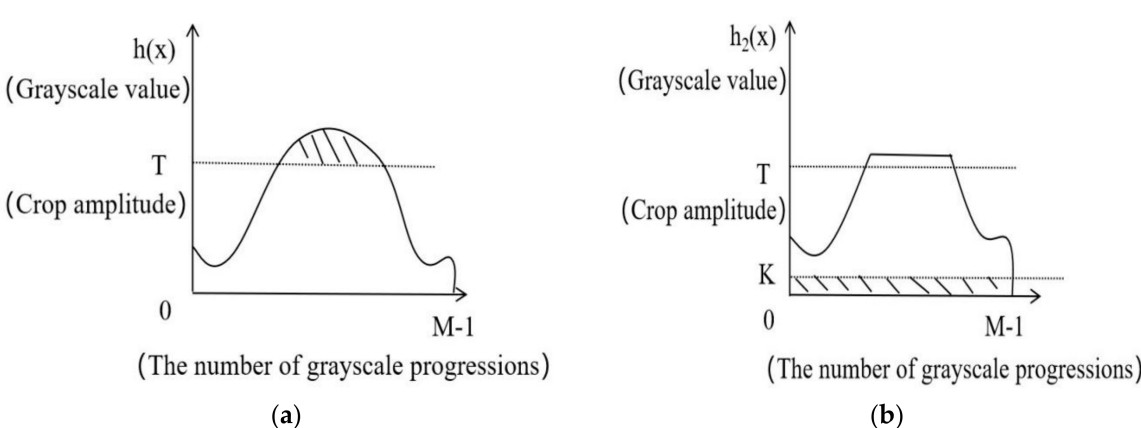

**Figure 5.** CLAHE histogram transformation process; (**a**) histogram; (**b**) restricted post-histogram.

Bilinear interpolation is a linear interpolation that occurs once in each row and column along the rows and columns of a weld image. The schematic is shown in Figure 6.

$$\begin{aligned} g(x, y) \approx &\frac{g(A)(x_2 - x)(y_2 - y)}{(x_2 - x_1)(y_2 - y_1)} + \frac{g(B)(x - x_1)(y_2 - y)}{(x_2 - x_1)(y_2 - y_1)} \\ &+ \frac{g(C)(x_2 - x)(y - y_1)}{(x_2 - x_1)(y_2 - y_1)} + \frac{g(D)(x - x_1)(y - y_1)}{(x_2 - x_1)(y_2 - y_1)} \end{aligned} \tag{10}$$

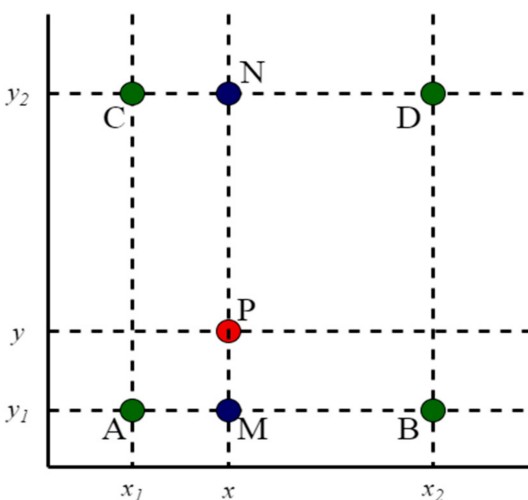

**Figure 6.** Bilinear interpolation schematic.

Among the known pixels are A $= (x_1, x_2)$, B $= (x_1, y_2)$, C $= (x_1, y_2)$, D $= (x_2, y_2)$. First, we calculate the insertion point M from the two known points A and B in the x-axis direction. In the same way we can calculate the insertion point N between the two points of C and D. We calculate the insertion point P by finding the known point M and N in the direction of the axis, and the P point is our final point to be inserted. By assuming the value of point P, the result of the interpolated point P is shown in Equation (1). Additionally, because A, B, C, and D are all adjacent points, the denominator $(x_2 - x_1)(y_2 - y_1) = 1$ in Equation (10) makes the algorithm faster.

(7) The gray value of the obtained image $H_1(x)$ after secondary noise reduction and the enhanced image $H_2(x)$ is linearly different, highlighting the detailed high-frequency information, and the enhanced image $H_3(x)$ is obtained.

(8) Finally, the resulting enhanced image $H_3(x)$ and $H_2(x)$ gray values are linearly superimposed to obtain the final enhanced image $H(x)$.

## 4. Experimental Results

In this paper, an experiment based on the improved CLAHE algorithm is performed using the horizontal gear robot welding of the anchor chain, as shown in the example in Figure 1. This section evaluates the quality of images from both subjective and objective evaluations. Subjective evaluation is the processed image through the direct judgment of the human eye. Objective evaluation is to evaluate the processed image through image quality evaluation indicators, including mean squared error MSE (mean square error), peak signal-to-noise ratio (PSNR), and structural similarity (SSIM), to judge the image display quality.

### 4.1. Subjective Evaluation

Using the proposed algorithm and bilateral filtering, CLAHE algorithms, literature [14] algorithms, literature [17] algorithms, etc., the anchor chain horizontal profile weld images are processed separately, and the processing results are shown in Figure 7 below.

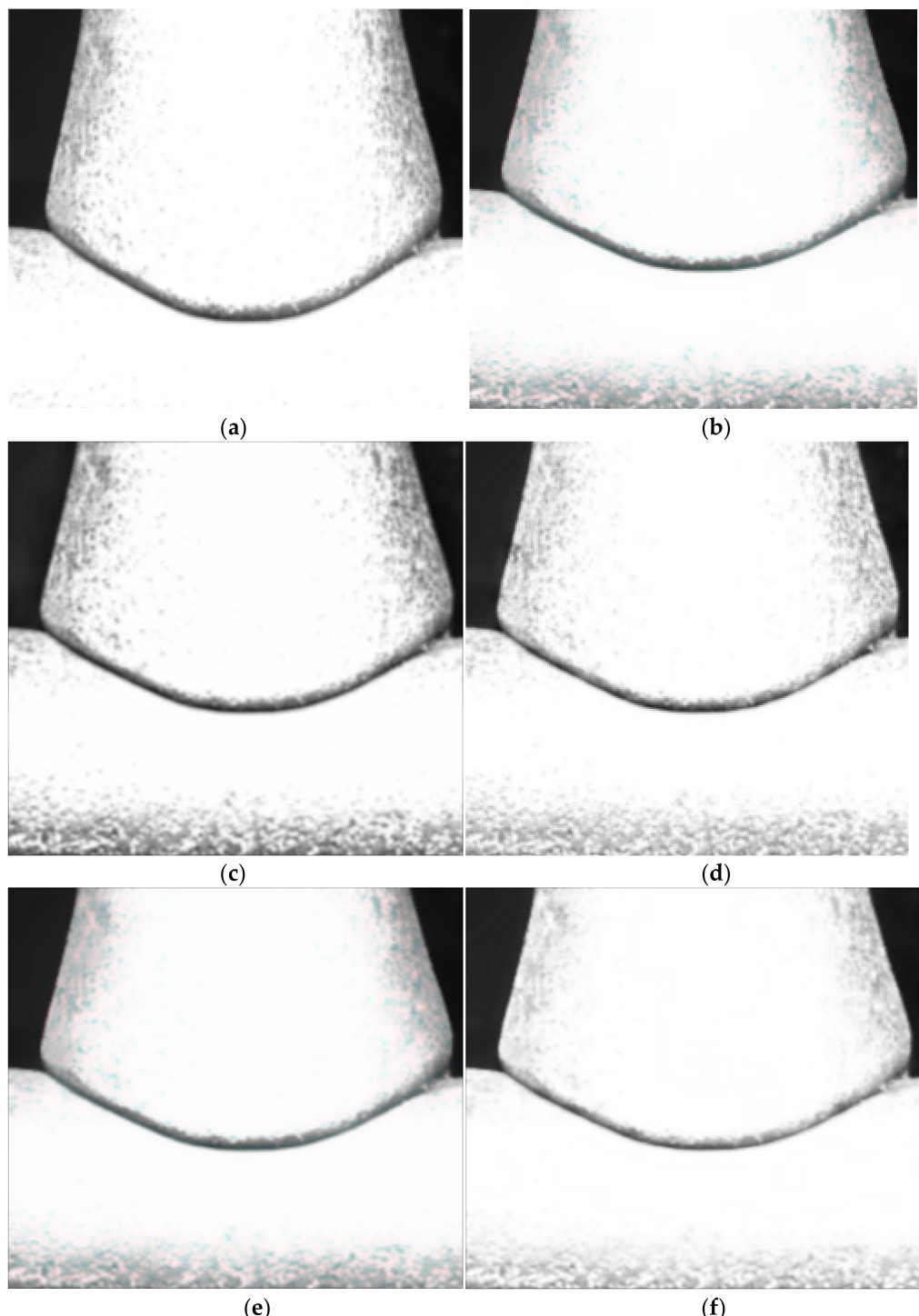

**Figure 7.** (**a**) Original image1, (**b**) bilateral filtering, (**c**) CLAHE, (**d**) literature [14] algorithms, (**e**) literature [15] algorithms, (**f**) this article's algorithm.

　　　As shown in Figure 7a, there are many noise points in the original figure, and there are irregularities in the edges of the weld. Figure 7b shows some improvement in image noise after bilateral filtering, but the edges of the weld are not smooth and clear. The CLAHE algorithm in Figure 7c has more noise points due to the large amount of noise present in the original image. In Figure 7d Document [14], the algorithm shows that the image noise point is slightly reduced, but the edges are still blurry. In Figure 7e Document [17], the algorithm shows that the image noise points are greatly improved, but the edges are not sufficiently clear. Figure 7f is visibly different to other algorithms; the image has relatively

fewer noise points, sharper and smoother edges, and higher image contrast. As can be seen from Figure 8f, the noise points of this algorithm are effectively improved compared to other algorithms, the edges of the weld seam have become smooth and clear, and the contrast of the image has been improved.

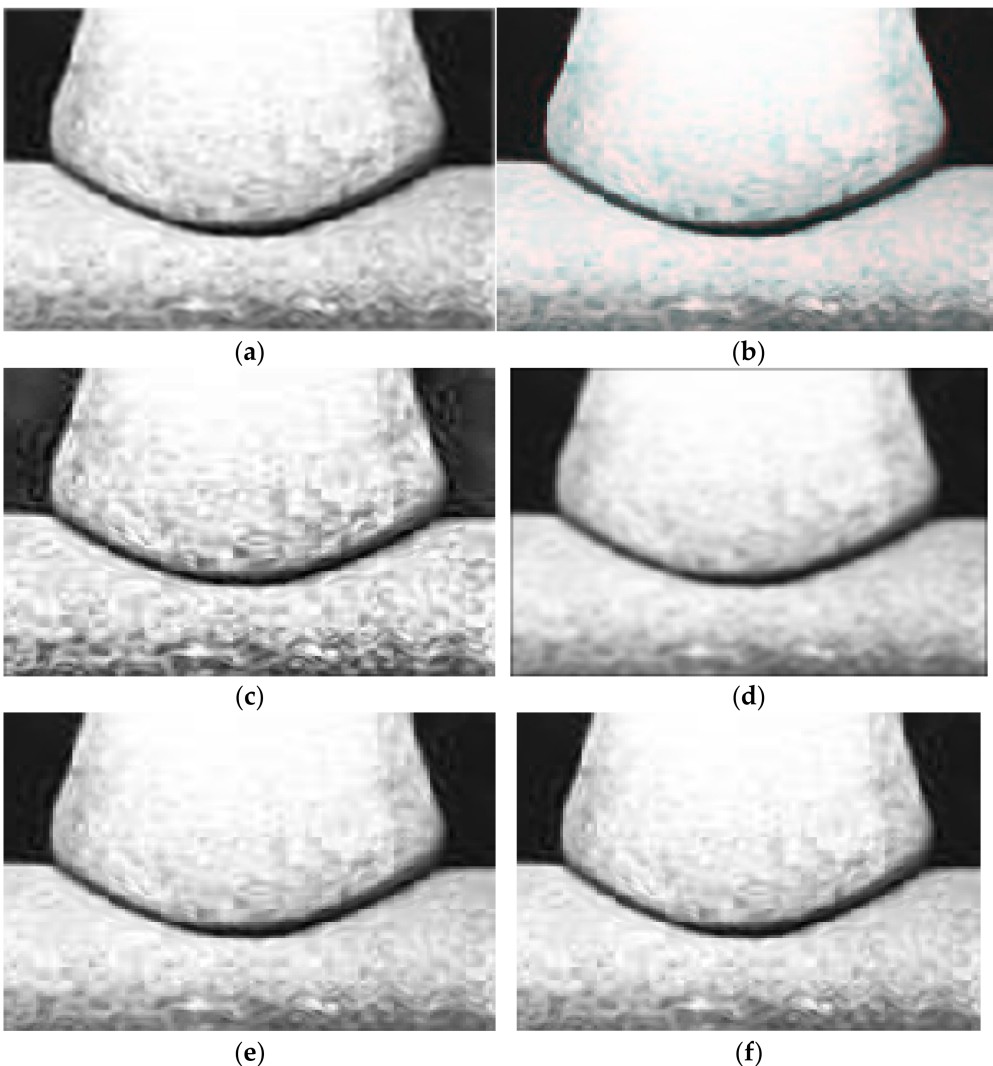

**Figure 8.** (**a**) Original image2, (**b**) bilateral filtering, (**c**) CLAHE, (**d**) literature [14] algorithms, (**e**) literature [17] algorithms, (**f**) this article's algorithm.

*4.2. Objective Evaluation*

4.2.1. Peak Signal-to-Noise Ratio (PSNR)

Peak signal-to-noise ratio is used to measure the image enhancement effect, the larger the peak signal-to-noise ratio; the more obvious the enhancement effect of the image, the higher the contrast. The peak signal-to-noise ratio formula is shown below:

$$MSE = \sum_i \sum_j |X(i,j) - Y(i,j)|^2 / N \tag{11}$$

$$PSNR = 10log_{10}((max(Y(i,j)))^2 / MSE \tag{12}$$

where *MSE* represents mean squared error, $X(I,j)$ represents the input image, $Y(i,j)$ represents the output image, $N$ represents the total number of pixels of the image, and $max(I(i,j))$ represents the maximum gray level of the output image.

4.2.2. Structural Similarity (SSIM)

Structural similarity is used to measure how similar the output image is to the input image; the closer the SSIM is to 1, the more similar the output image is to the input image, and the better the image processing effect. The structural similarity formula is shown below:

$$SSIM(X, Y) = \frac{(2\mu_X\mu_Y + C_1)(2\mu_{XY} + C_2)}{(\mu_X^2 + \mu_Y^2 + C_1)(\delta_X^2 + \delta_Y^2 + C_2)} \tag{13}$$

where $\mu_X$, $\mu_Y$, respectively, represent the gray value mean of the input image and the output image $Y$, $\delta_X$ represents the standard deviation of the input image $X$, $\delta_Y$ represents the standard deviation of the input image $Y$, and $\delta_{X,Y}$ is the root mean square of the covariance, and $C_1, C_2$ are constants.

In this paper, PSNR and SSIM are used to evaluate the algorithm, bilateral filtering, CLAHE algorithms, literature [14] algorithms, and literature [17] algorithms to process the anchor chain horizontal profile weld diagrams, respectively; the evaluation results are shown in Tables 1 and 2.

**Table 1.** PSNR values after different algorithms.

| The Kind of Image | Original Image 1 | Gain | Original Image 2 | Gain |
|---|---|---|---|---|
| Bilateral filtering | 27.584 | 5.57% | 24.806 | 3.05% |
| CLAHE algorithms | 24.378 | 16.7% | 21.227 | 17.04% |
| Literature [14] algorithms | 27.896 | 4.68% | 25.139 | 1.75% |
| Literature [17] algorithms | 28.154 | 3.8% | 25.324 | 1.03% |
| This article's algorithm | 29.267 | / | 25.587 | / |

**Table 2.** Structural similarity after enhancement by different algorithms.

| The Kind of Image | Original Image 1 | Gain | Original Image 2 | Gain |
|---|---|---|---|---|
| Bilateral filtering | 0.8839 | 1.9% | 0.7839 | 3.8% |
| CLAHE algorithms | 0.8439 | 6.78% | 0.7566 | 7.55% |
| Literature [14] algorithms | 0.8721 | 3.33% | 0.7986 | 1.89% |
| Literature [17] algorithms | 0.8979 | 0.36% | 0.7997 | 1.75% |
| This article's algorithm | 0.9011 | / | 0.8137 | / |

By comparing the peak signal-to-noise ratio and structural similarity of the anchor chain welding images, it can be seen from Table 1 that the peak signal-to-noise ratio of the proposed algorithm in the original image 1 is increased by 5.75%, 16.7%, 4.68%, and 3.8% respectively, compared with the bilateral filtering, CLAHE algorithm, literature [14] algorithm, and literature [17] algorithm. Compared with bilateral filtering, CLAHE algorithms, literature [14] algorithms, and literature [17] algorithms, the peak signal-to-noise ratio of the original image 2 was increased by 3.05%, 17.04%, 1.75%, and 1.03%, respectively. From the images and data, it can be found that after being enhanced by literature [13] algorithms, a large number of noise points appear in the image, and the image enhancement effect is not good. Through the enhancement of the algorithm, the weld image noise is effectively reduced, the weld edge is clearer, and the image contrast is improved. It can be seen from Table 2 that, compared with several other algorithms, the enhanced structural similarity of the algorithm is closer to 1, and compared with the literature [17] algorithm, the structural similarity of the original image 1 increases by 0.41%, and the structural similarity of the original image 2 increases by 1.72%, indicating that the algorithm in this paper retains more image details, the influence of noise is reduced during the image enhancement process, and the edge of the image weld becomes clearly visible.

## 5. Conclusions

In this paper, the image enhancement of the 1.25-megapixel weld image is performed by weld image processing of robot welds to improve the influence of weld blur edges and noise points. In this paper, an improved CLAHE algorithm is proposed, which first performs an improved bilateral filtering algorithm for the weld image to obtain the weld image after denoising and edge retention. Then, the processed image is enhanced by Gaussian mask processing and the CLAHE algorithm, respectively, and then the two algorithms are secondarily differentiated to obtain a detailed image with secondary noise reduction and image contrast improvement. After the improvement by the algorithm in this paper, the noise point of the weld image is slightly improved, and the edge of the weld becomes sharp and clear. In this paper, by simulating an anchor chain weld image, it is found that compared with other algorithms, the image quality evaluation index PSNR is increased by about 3.8~4.68%, the SSIM is close to 1, the noise point of the image is reduced, the clarity of the contour edge is improved, the contrast of the image is also improved, and the visual effect is further improved, thus proving the effectiveness of the algorithm proposed in this paper.

Subsequently, the image will be enlarged by a higher magnification and the enhancement of the weld edge feature information will be identified.

**Author Contributions:** Conceptualization, P.L.; methodology, P.L.; software, P.L.; validation, P.L.; formal analysis, P.L.; investigation, P.L.; resources, P.L.; data curation, P.L.; writing—original draft preparation, P.L.; writing—review and editing, P.L. and Q.H.; visualization, P.L. and Q.H.; supervision, P.L.; project administration, P.L.; funding acquisition, P.L. All authors have read and agreed to the published version of the manuscript.

**Funding:** This research received no external funding.

**Data Availability Statement:** Data are contained within the article.

**Conflicts of Interest:** The authors declare no conflict of interest. The funders had no role in the design of the study; in the collection, analyses, or interpretation of data; in the writing of the manuscript; or in the decision to publish the results.

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
