# Peer review of "Robotic Weld Image Enhancement Based on Improved Bilateral Filtering and CLAHE Algorithm"

_electronics, doi:10.3390/electronics11213629_

Round 1

Reviewer 1 Report

The paper looks fine. For improvement please add pseudo code for the algorithm. Compare this to the work done by other researchers. Also, please highlight the contribution of this paper with originality. It is better if the images are of same size. 

Reviewer 2 Report

Dear Authors, 

I think your manuscript is well written, original and ready to be published, but not in this journal.

I think that your work does not fit with the aims and the scope of this journal. For this reason, I suggest you finding a better journal for your manuscript. However, I leave to the editor the responsibility of accepting or not the manuscript.

I have just 1 minor comment:

1) You should explain better eq. 1-2-3 and be sure that each letter identifies only one function. It is difficult to understand what the difference between I(i,j) and I(x,y) is in your paper. 

Regards

Reviewer 3 Report

An interesting application of current methods of Computer Vision (namely the CLAHE parametric histogram equalization) to a real-world use case of the analysis of weld joins. Nicely introduced use case, pretty decent language, some illustratory photos. In the middle of the paper the authors struggle to include some math, which is not very useful for the paper, since the math presented include something useful only in the equation (10), so in my opinion the paper would be more practical if some math would be skipped. If it is applied science, it should be written to make it work. If it is theoretical research, then, well, it is not strong, let's say. So let's go towards the applied science scenario.

I do not like the Conclusions chapter.

It says "enhanced", "differentiated", "improvement", "significantly reduced", "sharper and clearer", "visibly enhanced", "quality increased", "noise greatly reduced", "outline clearer", "further enhanced", etc. --- while --- I do not see the Figure8(f) to be "so much better" than [14] or [15]. To be honest, for sharp edges and lower noise I would pick the Fig8(b) rather than (f). So I do not "feel" the narrative of Conclusions.

Of course, the authors have done some work implementing all those methods (even if OpenCV makes it easier nowadays), so a lot of (/some) work has been done, so the paper is worth publishing (in the applied science context), but the narrative of the Conclusion should be more diplomatic. It is not a sin to make a new approach which is SOMETIMES better -or- for SOME cases -or- it IS better, although ONLY BY 4.68%. That is great. You do not have to find a method better by 200%.

In my opinion the narrative of Conclusions should be more realistic and less "emphasized" (exaggerated?), and then the paper will be good enough.

//+technical formatting (commas, spaces, image/eq placement, etc.)

Reviewer 4 Report

Overall good effort; if all the mentioned results in percentages could be put on a graph/chart, it would be clearer to understand.

Round 2

Reviewer 2 Report

Dear Authors, 

I am satisfied with your answer.

Sincerely.